# New Formulation to Synthetize Semiconductor Bi$_2$S$_3$ Thin Films Using Chemical Bath Deposition for Optoelectronic Applications

**Amanda Carrillo-Castillo** [1,*] , **Brayan G. Rivas-Valles** [1], **Santos Jesus Castillo** [2], **Marcela Mireles Ramirez** [3] **and Priscy Alfredo Luque-Morales** [4]

1 Institute of Engineering and Technology, Autonomous University of Ciudad Juarez, Ciudad Juárez C.P. 32310, Chihuahua, Mexico
2 Physics Research Department, University of Sonora, Hermosillo C.P. 83000, Sonora, Mexico
3 Department of Electrical Engineering, University of Texas at Dallas, Richardson, TX 75080, USA
4 Faculty of Engineering, Architecture and Design, Autonomous University of Baja California, Ensenada C.P. 22860, Baja California, Mexico
* Correspondence: amanda.carrillo@uacj.mx

**Abstract:** Anisotropic materials possess direction dependent properties as a result of symmetry within their structure. Bismuth sulfide (Bi$_2$S$_3$) is an important semiconductor exhibiting anisotropy due to its crystalline and stratified structure. In this manuscript we present a new and straightforward procedure to deposit Bi$_2$S$_3$ thin films on soda lime glass substrates by the chemical bath deposition (CBD) technique. We studied two fundamental parameters, the time to deposit a single layer and the total number of layers deposited. The single layer deposition time was varied between 70 and 100 min and samples were coated with a total of 1, 2, or 3 layers. It is important to note that a fresh aqueous solution was used for every layer. Visible and near infra-red spectroscopy, scanning electron microscopy, X-ray photoelectrons spectroscopy, and X-ray diffraction were the characterization techniques used to study the resulting films. The calculated band gap values were found to be between 1.56 and 2.1 eV. The resulting Bi$_2$S$_3$ deposited films with the new formulation showed uniform morphology and orthorhombic crystalline structure with an average crystallite size of 19 nm. The thickness of the films varied from 190 to 600 nm in direct correlation to the deposition time and in agreement with the number of layers. The XPS results showed the characteristic bismuth doublet centered around 164.11 and 158.8 eV corresponding with the presence of Bi$_2$S$_3$. The symmetry within the Bi$_2$S$_3$ structure makes it a strong anisotropic crystal with potential applications in optoelectronic and photovoltaic devices, catalysis, and photoconductors among others.

**Keywords:** Bi$_2$S$_3$; chemical bath deposition; thin films; semiconductor

## 1. Introduction

Semiconductor thin films have been the basis of the development of an astounding variety of modern technologies. In recent years, research in this field has been accelerated by the realization and success of several applications [1–5]. A thin film is a layer of material, which can be an insulator, a semiconductor, or a conductor, with a thickness from a couple to hundreds of nanometers. Thin films have enabled countless applications such as electronic devices, transistors, solar cells, solid state illumination, sensors, and data storage [6]. To date, new applications are still being developed, even for the medical field such as biosensors and scaffolds for tissue engineering [7–10].

Thin films can be obtained through a vast catalog of methods; some of the most used are spray pyrolysis, physical or chemical vapor deposition, electrochemical deposition, successive ionic layer adsorption/reaction, and chemical bath deposition [11–16].

Bismuth based materials exhibit unique structure, high carrier mobility, and environmental stability, which makes them attractive for optoelectronic applications [17]. Moreover,

bismuth is known for its significant infrared refractive index ($n{\sim}10$) as well as negative ultraviolet-visible permittivity [18].

Bismuth chalcogenide has demonstrated a stability in ambient conditions and can be easily integrated into electronic devices fabricated on traditional rigid or flexible substrates. In addition, its semiconducting properties can be varied with the addition of oxygen [19,20].

Bismuth sulfide ($Bi_2S_3$) is a binary salt also called bismuthinite and it is the most common form in which bismuth is found in nature. Bismuthinite is cataloged as a chalcogenide semiconductor. $Bi_2S_3$ thin films have been obtained using chemical and physical methods, aerosol assisted chemical vapor deposition (CVD) [14], melt-quench techniques [21], successive ionic layer adsorption and reaction (SILAR) methods [22], and chemical bath deposition (CBD) [23,24]. CBD is perhaps the simplest and most affordable method for depositing thin films. It is based on an aqueous solution and a substrate in which the material is to be deposited. Film growth during CBD can occur ion-by-ion or though the formation of a hydroxide cluster. In the first mechanism, sequential ionic reactions are involved; the second mechanism requires the formation of a metal complex, which also prevents precipitation. In this work, film growth followed the second mechanism since $Bi(OH)_3$ was formed first (see reaction mechanism proposed). The initial nucleation of hydroxide occurs homogeneously in solution; therefore, the chalcogenide is formed homogeneously as well and usually precipitates out of solution. It is advisable to control the rate of the chemical reactions involved, to remain slow enough as to allow the chalcogenide to gradually form on the substrate as well as to diffuse and adhere either to the substrate itself or to the growing film [25]. This simple system restricts some of the variables that would allow controlling some of the properties of the resulting thin film, including temperature stability, deposition angle, and precursor solubility [1,3,9].

Different authors have utilized $Bi_2S_3$ as an *n*-type semiconductor with a 1.6–1.9 eV band gap [1,3,26]. The properties of semiconductor materials can be influenced by the symmetry of their structure. The crystal structure of $Bi_2S_3$ is strongly anisotropic and stratified, presenting as orthorhombic bismuthinite. The latter makes this material a good candidate for optoelectronic applications [27–29]. Unlike other materials used for solar cells, its toxicity is considerably low. Depositing this material, as a thin film for its application in electronic devices, stiff or flexible, has not been explored methodically. For these reasons, the development of a low-cost route for its synthesis is of high interest.

This work focuses on a new formulation for the synthesis of $Bi_2S_3$ thin films, which were characterized by visible and near infra-red (Vis-NIR) spectroscopy to analyze their absorption response properties, scanning electrons microscopy (SEM) to determine their morphology and thicknesses, X-ray photoelectron spectroscopy to confirm the chemical composition, and X-ray diffraction (XRD) to investigate the crystalline structure and crystallite size.

## 2. Materials and Methods

The $Bi_2S_3$ thin coatings were deposited on soda lime glass substrates (25 × 75 mm), these glass substrates were cleaned in an ultrasonic bath with acetone followed by isopropanol and finally rinsed with distilled water and dried with $N_2$ gas.

The used reagents were: Pentahydrate bismuth nitrate $Bi(NO_3)_3 \cdot 5H_2O$ (ACS reagent, ≥98.0%, ALDRICH), triethanolamine $C_6H_{15}NO_3$ (TEA) (99%, J.T. Baker), sodium hydroxide NaOH (ACS Reagent Grade 97%, FERMONT), and thiourea $CH_4N_2S$ (ACS reagent, ≥98.0%, J.T. Baker).

Our recipe designed to deposit $Bi_2S_3$ consists of a mixture of 5 mL of TEA (1 M) with 40 mL of $Bi(NO_3)_3 \cdot 5H_2O$ (0.1 M), 2.5 mL of TEA ($C_6H_{15}NO_3$) (0.5 M), 2.5 mL of sodium hydroxide (NaOH) (1 M), and 5 mL of thiourea ($CH_4N_2S$) (0.15 M). The role of each reagent is as follows: to provide bismuth ions, to act as a complexing agent, to raise the pH, and to provide sulfur ions, respectively. The reagents were added to the beaker in the order listed, and the beaker was stirred after each addition to ensure a homogeneous solution. The solution was poured into a 50 mL beaker where 40 mL of water was added. The thermal

bath temperature was chosen to be 60 °C and the beaker was placed in the bath promptly after all the reagents were added, the pH of the solution was kept at a value of 12. After deposition, the $Bi_2S_3$ films were cleaned in an ultrasonic bath with methanol followed by distilled water and dried with $N_2$ gas.

Initially, we studied reaction times from 10 to 180 min and found 70, 80, 90, and 100 min to be the most stable, resulting in homogeneous coatings and improving film adhesion. Then, we evaluated the effect of the number of layers (1, 2, and 3), and selected reaction or deposition times (70, 80, 90, and 100 min). As we are mainly interested in developing a new synthesis route, we propose the following reaction mechanism:

$$Bi(NO_3)_3 \cdot 5H_2O + 3Na(OH) \rightarrow Bi(OH)_3 + 3Na(NO_3) + 5H_2O \tag{1}$$

$$2Bi(OH)_3 + 2n[C_6H_{15}NO_3] \rightarrow 2[Bi(TEA)n]^{3+} + 6(OH)^{1-} \tag{2}$$

$$SC(NH_2)_2 + 2H_2O \rightarrow H_2S(g) + CO_2(g) \uparrow + 2NH_3(g)\uparrow \rightarrow S^{2-} + H_2O \tag{3}$$

$$[Bi(TEA)n]^{3+} \rightarrow Bi^{3+} + n(TEA) \tag{4}$$

$$2\,Bi^{3+} + 3S^{2-} \rightarrow Bi_2S_3 \tag{5}$$

We deposited twelve $Bi_2S_3$ samples, identified with labels following the nomenclature "Smn" explained in Table 1.

**Table 1.** $Bi_2S_3$ samples deposited by chemical bath deposition.

| Sm,n~Total Deposition Time (Minutes) | m, Represents the Number of Layers | | |
|---|---|---|---|
| n, represents the deposition time | $S_{1,70}$~70 | $S_{2,70}$~140 | $S_{3,70}$~210 |
| | $S_{1,80}$~80 | $S_{2,80}$~160 | $S_{3,80}$~240 |
| | $S_{1,90}$~90 | $S_{2,90}$~180 | $S_{3,90}$~270 |
| | $S_{1,100}$~100 | $S_{2,100}$~200 | $S_{3,100}$~300 |

Our characterization included UV-Vis-(NIR), SEM, XPS, and XRD. The UV-Vis-(NIR) measurements were carried out with a Jenway 6850 with a 0.1 nm resolution in the 300–1100 nm range. The SEM images were obtained with a JEOL 7000 F JSM using an acceleration voltage of 0.1–30 KV. The average $Bi_2S_3$ films thicknesses were obtained by measuring the SEM cross-section. Chemical analysis was done with an X-ray photoelectron spectrometer from Perkin Elmer Phi 5600 ESCA system equipped with an Aluminum (Al) X-ray source. The crystalline structure of the $Bi_2S_3$ films was investigated in a Rigaku Ultima III X-ray diffractometer with $CuK\alpha$ $(\lambda)$ = 1.54 Å, operated at 40 kV and 44 mA with a scan step set at 0.5 °/min.

## 3. Results and Discussion

We first characterized the absorption of the films between 350 and 1100 nm. Figure 1 presents the absorption of the three sets of samples corresponding to films deposited with different number of layers and deposition times. The Tauc method was used to calculate the direct band gap value from the absorbance spectra (Figure 2).

From the Tauc method $\frac{d^2\tau}{dE^2} \equiv 0$, assuming a direct band gap $\tau = [(A)(E)]^2 = C(E - E_g)$, where $\tau$ = Tauc variable, C = slope of linear behavior, $E_g$ = Energy band gap, E = Incident energy, and A = A(E) = Absorption of the coating. $\frac{d^2\tau}{dE^2} = \frac{d^2C(E-E_g)}{dE^2} \equiv 0$

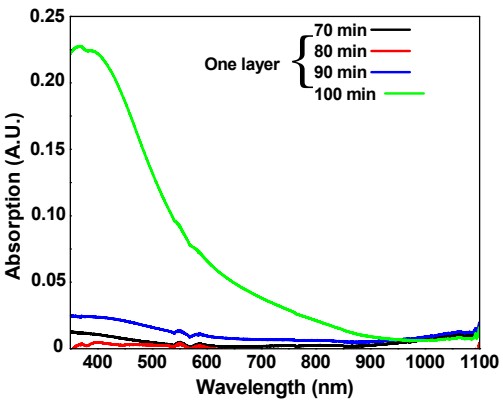

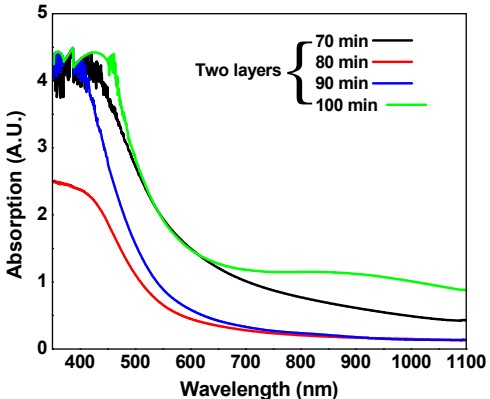

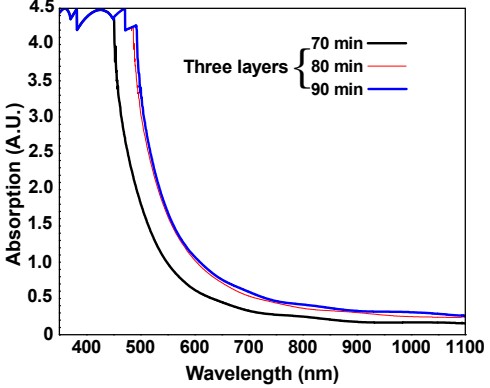

**Figure 1.** Obtained absorption spectra of different as ground $Bi_2S_3$ films, labeled as it is indicated in figure.

The $Bi_2S_3$ samples showed to have an absorption edge within 950–1100 nm depending on deposition time and the number of layers, this corresponds to 2.1–1.6 eV band gap (Figure 3). The optical band gap (Eg) decreased when the number of layers increased. A decrease of the band gap can be attributed to an increase of the grain size of the $Bi_2S_3$ films. The effect of grain size on the optical band gap can be attributed to quantum confinement effects [30–34]. The obtained values in this work are in agreement with the ones reported by others [1,3,25]. In the range from 700 to 1100 nm, the absorption intensity can be correlated to the increase of the deposition time for samples consisting of one and three layers. The reason for the nonlinear tendency in absorption intensity for the samples with two layers may be the presence of structural defects.

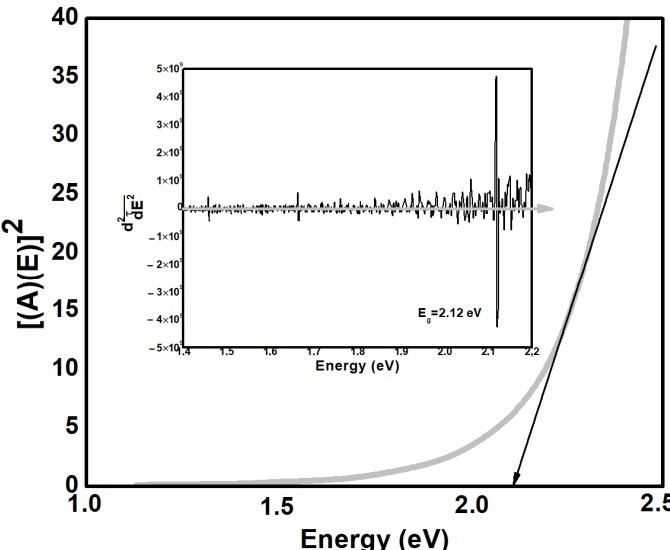

**Figure 2.** Tauc variable vs. Energy plot for selected bismuth sulfide film deposited by CBD.

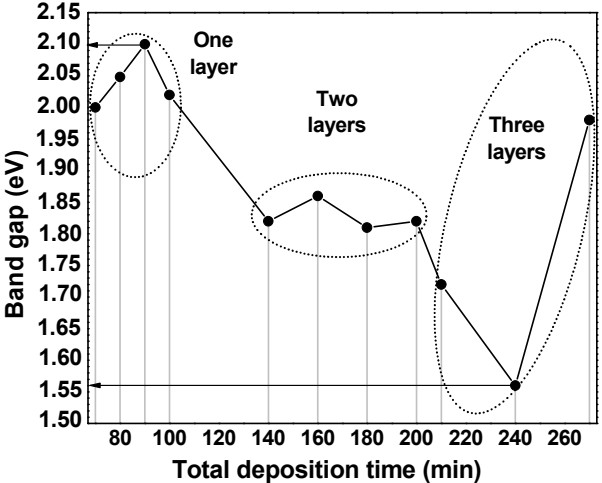

**Figure 3.** Direct band gap obtained for each subgroup of $Bi_2S_3$ films as function of total deposition time.

Figures 4–6 show the morphology of the films with one, two, and three layers. A level of homogeneity was observed for the following samples: one layer—-100 min; two layers—-70, 80, 90, and 100 min; and three layers—-70, 80, and 90 min.

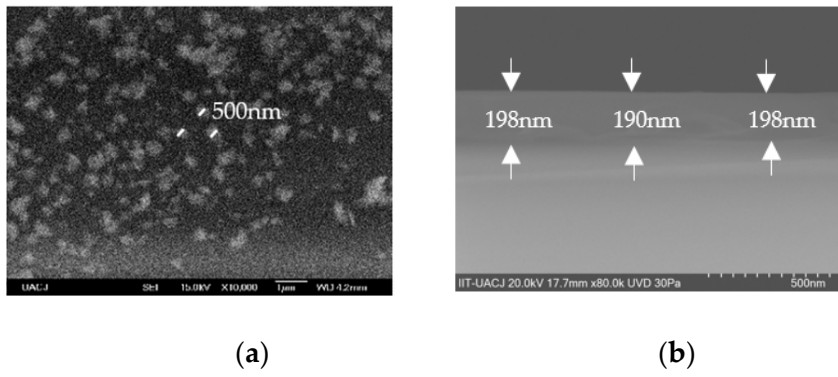

(**a**)          (**b**)

**Figure 4.** SEM micrographs for $Bi_2S_3$ samples containing one layer grown with a deposition time of 100 min (**a**) film morphology (**b**) SEM image cross-section.

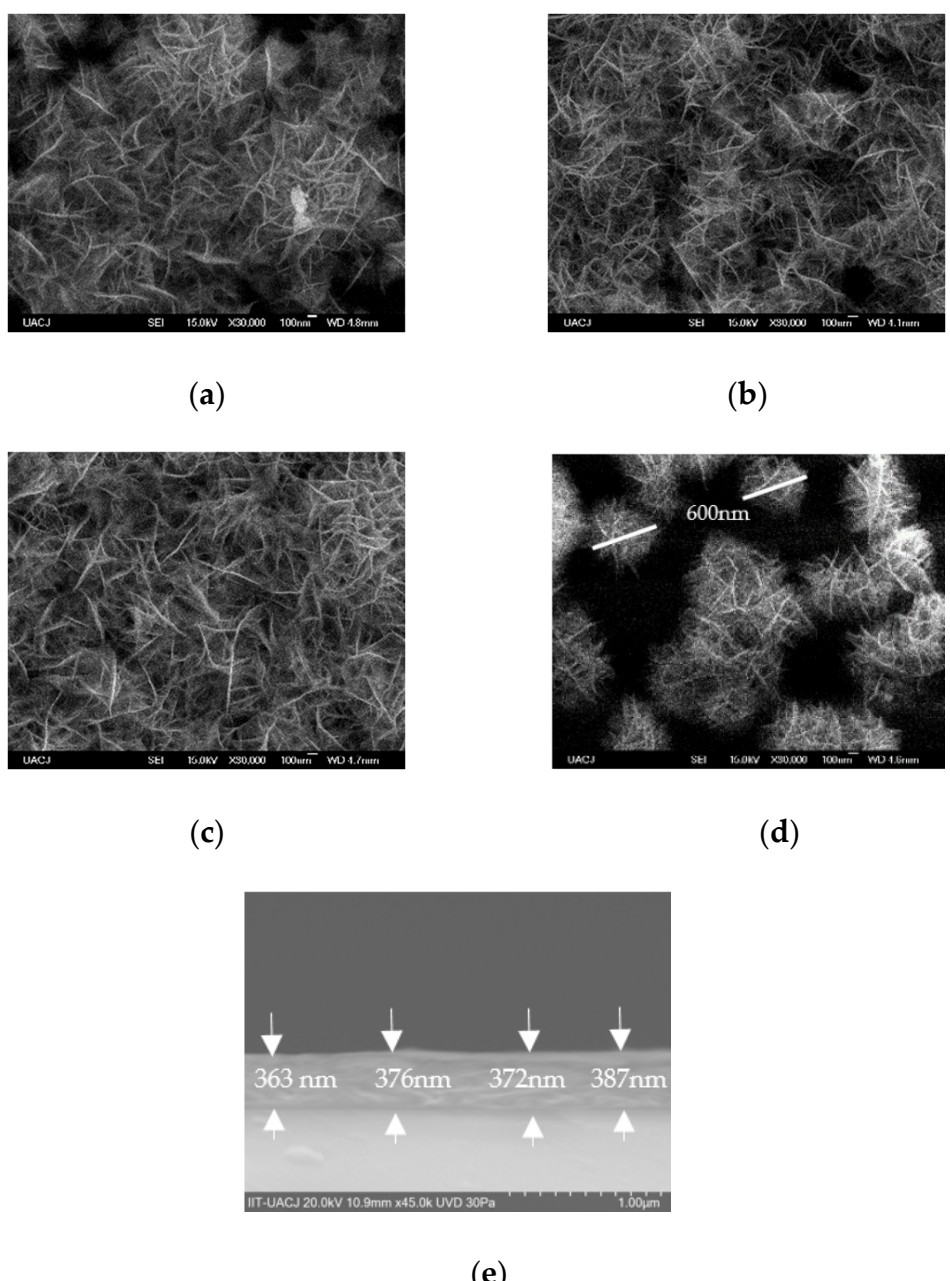

**Figure 5.** SEM micrographs for $Bi_2S_3$ samples deposited with two layers grown with a deposition time of: (**a**) 70 min, (**b**,**e**) 80 min, film morphology and SEM image cross-section, respectively, (**c**) 90 min, and (**d**) 100 min.

Figure 4a shows the morphology of the film with one layer and 100 min deposition time; the morphology shows aggregates resembling isolated sea urchins with a size of 500 nm–1 μm; the size increases with the number of layers. Figure 4b shows a SEM image cross-section for the film where a thickness of ~195 nm on average was measured.

Figure 5 shows the morphology for the samples deposited with two layers and different deposition time.

The surface morphology resembles entangled sea urchins, as can be observed in parts a, b, c, and d of Figure 5. However, a bilayer formed when using a deposition time of 100 min showing larger sea urchins that are better defined; this deposition time was discarded and no longer pursued when studying three-layer samples.

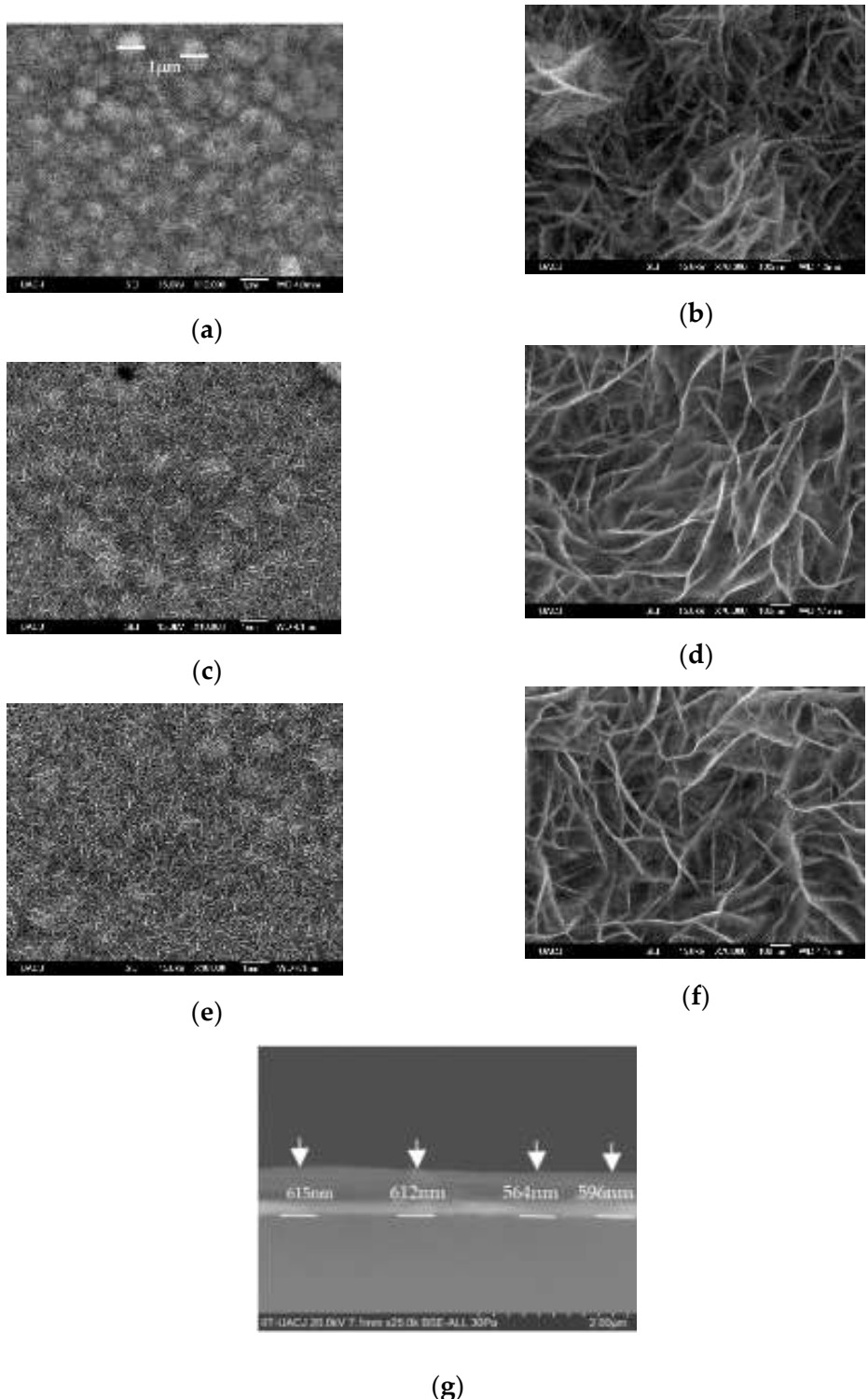

**Figure 6.** SEM morphology of three Bi$_2$S$_3$ three layers grown with a deposition time of: (**a**,**b**) 70 min, (**c**,**d**,**g**) 80 min, film morphology (two micrographs), and SEM image cross-section, respectively, and (**e**,**f**) 90 min.

Figure 6 shows the morphology of the obtained samples deposited with three layers. Overall, the films seem to follow the same trend as the films deposited with two layers.

The long deposition time for the growth of Bi$_2$S$_3$ thin films by CBD might be due to the use of stronger complexing agents in the process, as in some cases becomes necessary to assist the dissolution of the bismuth salt with agents such as TEA.

The SEM images showed an incomplete coverage of material over the surface of the substrates for $Bi_2S_3$ films deposited with one layer; however, formation of more homogeneous films was achieved upon increasing the number of layers and deposition. This is due to the fact that growth on a substrate depends on the formation of nucleation sites that form during the deposition of the first layer. The process of aggregation then occurs during subsequent layers. High concentrations of bismuth or sulphur can produce a high number of particles in each layer. In theory, the van der walls forces increases if two particles approach each other, until an individual particle has formed [25]. SEM micrographs of $Bi_2S_3$ films deposited with one, two, and three layers demonstrate the process of aggregation during the growth of thin films, similar morphology has been reported elsewhere [31,32]. The morphology of $Bi_2S_3$ films could be attributed to the film growth proceeding cluster-by-cluster, as proposed in this work. The latter can strongly depend on the complexing agent, which acts by suppressing the growth of certain crystal facets via coordination to metal cations, in this case with the use of a stronger complexing agent [32,33].

In the present work, we report on the preparation of $Bi_2S_3$ homogeneous and compact microstructure films. In the next section, we focus on the $Bi_2S_3$ films deposited with two and three layers and a deposition time of 80 min.

In order to confirm the presence of $Bi_2S_3$, XPS was carried out after mild surface cleaning with an Ar sputter gun to remove the adsorbed species from the atmosphere. Figure 7 shows the results obtained for the Bi 4f region; the data have been fitted with a doublet showing an energy splitting of 5.31 eV. The main peak in the doublet, Bi 4f 7/2, was found to be centered at 158.8 eV, which is in agreement with the presence of $Bi_2S_3$ [35].

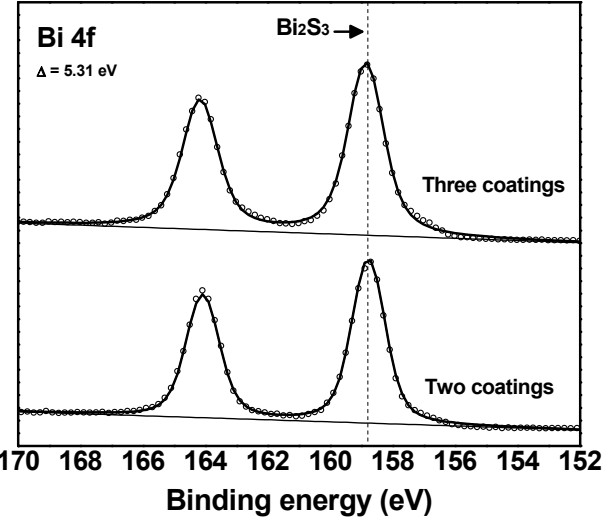

**Figure 7.** Spectra from X-ray photoelectron spectroscopy for the Bi 4f region taken from the $Bi_2S_3$ thin films analyzed.

Figure 8 shows the XRD pattern obtained for a $Bi_2S_3$ sample deposited with three layers and a deposition time of 80 min. The crystalline planes are indexed from reference to PDF 17-0320 for $Bi_2S_3$-bismuthinite [1–4,8,9]. The average crystallite diameter for $Bi_2S_3$ was calculated to be 19.07 nm, as estimated from the (221) peak. In general, the resulting properties such as the band gap and crystalline structure are similar to previous reports. The samples in this work did not require any post-deposition processing to obtain $Bi_2S_3$ films. The reaction mechanism reported here is a new contribution and agrees well with the different deposition times and number of layers deposited [36–38].

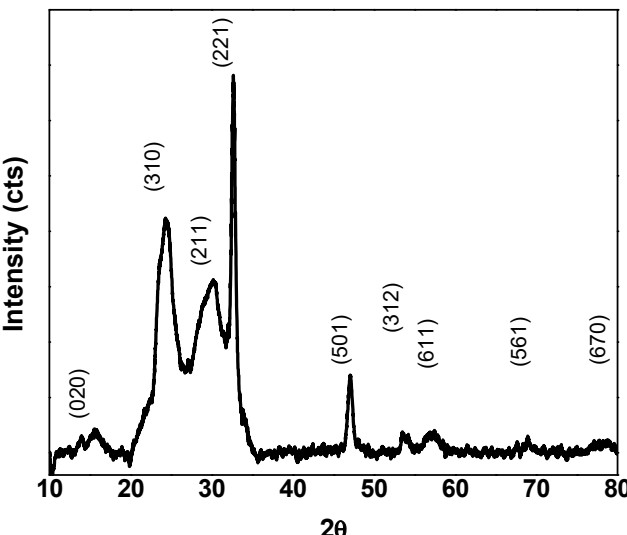

**Figure 8.** XRD patterns of $Bi_2S_3$ films deposited with three layers and 80 min deposition time.

## 4. Conclusions

The present work shows the effect of the deposition time and number of layers on $Bi_2S_3$ films obtained by chemical bath deposition. The methodology presented here yields uniform films deposited at a low temperature. These processing conditions play an important role in the morphology and optical properties. The XPS characterization demonstrated the presence of $Bi_2S_3$ and the diffraction data showed the presence of its orthorhombic phase. The low temperature and low-cost process to obtain $Bi_2S_3$ films presented here complies with the requirements for optoelectronic and flexible electronics technologies and can be extended to the fabrication of semiconductor devices.

**Author Contributions:** A.C.-C.: Conceptualization, Methodology, Validation, Formal analysis, Investigation, Resources, Writing—Original Draft, Writing—Review and Editing, Visualization, Project administration, Funding acquisition. B.G.R.-V.: Methodology, Writing—Review and Editing Marcela Mireles: Validation, Writing—Review and Editing, Investigation, Visualization. S.J.C.: Conceptualization, Methodology, Validation, Formal analysis, Investigation, Review and Editing, Visualization. M.M.R.: Validation, Writing—Review and Editing, Investigation, Visualization. P.A.L.-M.: Validation, Writing—Review and Editing, Investigation, Visualization. All authors have read and agreed to the published version of the manuscript.

**Funding:** The authors acknowledge partial financial support from CONACyT through the grants Problemas Nacionales 3529-2016 and Ciencia Básica 2013-IOO17-22111.

**Data Availability Statement:** Not applicable.

**Acknowledgments:** The authors acknowledge partial financial support from CONACyT through the grants Problemas Nacionales 3529-2016 and Ciencia Básica 2013-IOO17-22111, Government of the State of Chihuahua through the Secretariat of Innovation and Economic De-velopment, through the Institute of Innovation and Competitiveness and the staff of the flexible electronics laboratory of UACJ.

**Conflicts of Interest:** The authors declare no conflict of interest.

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
