# Peer review of "New Formulation to Synthetize Semiconductor Bi2S3 Thin Films Using Chemical Bath Deposition for Optoelectronic Applications"

_symmetry, doi:10.3390/sym14122487_

Round 1

Reviewer 1 Report

1.  The authors used myriad typos and grammatical and scientific mistakes in the whole manuscript. It should have been read and edited by a native English speaker with knowledge of the techniques. In addition, there are some subscripts and superscripts editing issues. 

2. Why Bi2S3 thin films are important as compared to other materials like Bi2Se3, Bi2O3, Bi2Te3, and several other classes?

3.  Why is the quality of SEM images awful? The text written in the images is unclear. Please provide high-quality images.

4.  Authors mentioned in the summary that their growth strategy could be used for optoelectronics. Authors must explain how their growth mechanism is better than CVD-grown sample, which provides ultra-clean, thin and large domain samples. Please note that the quality of the material is more important than the growth mechanism.

5. Please index peaks around 69 and 78 in the XRD pattern.

6. Authors must explain the growth methodology for achieving the flower-shaped nanostructures. Please provide a detailed growth methodology.

7. Also, please find the thickness of flakes associated with each particle. Such structures are useful for electrocatalysis, so you provide the necessary information for their disciplines. 

Author Response

Response to referees for Ms. Ref. No.:  1936969

Dear Reviewers: We sincerely thank the reviewers for the time invested in reviewing our paper as well as their valuable comments. We have revised the paper taking into account the comments provided. We are now submitting the revised manuscript for review and potential publication.

Response to reviewer #1

1. The authors used myriad typos and grammatical and scientific mistakes in the whole manuscript. It should have been read and edited by a native English speaker with knowledge of the techniques. In addition, there are some subscripts and superscripts editing issues.

RE: Done. We have revised and corrected grammatical and scientific mistakes.

2. Why Bi2S3 thin films are important as compared to other materials like Bi2Se3, Bi2O3, Bi2Te3, and several other classes?

RE: Done. Per the reviewer suggestion, we have added more literature in Introduction and Results section about growth mechanism by CBD and chalcogenide semiconductors.

3. Why is the quality of SEM images awful? The text written in the images is unclear. Please provide high-quality images.

RE: Done. Per the reviewer suggestion, we have improved the SEM images and the text in the graphics.

4. Authors mentioned in the summary that their growth strategy could be used for optoelectronics. Authors must explain how their growth mechanism is better than CVD-grown sample, which provides ultra-clean, thin and large domain samples. Please note that the quality of the material is more important than the growth mechanism.

RE: Done. Per the reviewer suggestion, we have added more literature in Introduction and Results section about growth mechanism by CBD, chalcogenide semiconductors and chalcogenide semiconductors applied in optoelectronic devices, specifically Bi2S3. We have added actualized references.

5. Please index peaks around 69 and 78 in the XRD pattern.

RE: Done.   

6. Authors must explain the growth methodology for achieving the flower-shaped nanostructures. Please provide a detailed growth methodology.

RE: Done. Per the reviewer suggestion, we have added more literature in Introduction and Results section about growth mechanism by CBD and we compared our results with other works.

7. Also, please find the thickness of flakes associated with each particle. Such structures are useful for electrocatalysis, so you provide the necessary information for their disciplines.

RE: Done.

Additionally all sections were restructured according reviewers comments and we actualized the references that give support our research.

Note: Changes to the manuscript have been marked in yellow.

Author Response

Response to referees for Ms. Ref. No.:  1936969

Dear Reviewers: We sincerely thank the reviewers for the time invested in reviewing our paper as well as their valuable comments. We have revised the paper taking into account the comments provided. We are now submitting the revised manuscript for review and potential publication.

Response to reviewer #2

  1. It is not clear what is the difference between them and the present manuscript. Moreover there is no any description of the strategy of the preparation method and what are the expected benefits. For these reasons I consider that the novelty of the manuscript is limited and I propose the rejection of the manuscript except if the authors can be prove that significant novelty included and the results were compared with previous works.

RE: Done. Per the reviewer suggestion, we have added more literature in Introduction and Results section about growth mechanism by CBD and we compared our results with other works.

  1. Also there are some minor mistakes.

RE: Done. We have revised and corrected grammatical and scientific mistakes.

  1. The reactions at lines 89-91 must be corrected (as it is a water solution use the ionic forms)

RE: Per the reviewer suggestion, we have added more literature in Introduction and Results section about growth mechanism by CBD. In this way we proposed the principal growth mechanism in our research be cluster by cluster,  for this reason we considerate correct our mechanism proposed supported by all characterization of thin films grown at different time and different number of chemical bath deposition. (line 226)

  1. In figure 1 Label and in lines 128, 179, replace Bi2S3 with Bi2S3

RE: Done. We have revised and corrected grammatical and scientific mistakes.

  1. At the end of line 133 remove the “is”

RE: Done. We have revised and corrected grammatical and scientific mistakes and we have restructured some discussions.

  1. At figure 3 the loops are not noticeable please use more intense ink. Also please explain why

RE: Done. We have revised and corrected grammatical and scientific mistakes and we have restructured some discussions.

  1. The last of the three coatings has a much bigger Eg.

RE: Done. Per the reviewer suggestion, we have improved the text in the graphics.

  1. The figure 4 is not clear please use bigger or better

RE: Done. Per the reviewer suggestion, we have improved the SEM images and the text in the graphics.

All sections were restructured according reviewers comments and we actualized the references that give support our research.

Note: Changes to the manuscript have been marked in yellow.

Reviewer 3 Report

The results are of potential interest to related readers. The manuscript is recommended for publication after a major revision.

(a) A thorough revision in English is necessary.

(b) Most of the figures should be redrawn by enlarging labels.

Author Response

Response to referees for Ms. Ref. No.:  1936969

Dear Reviewers: We sincerely thank the reviewers for the time invested in reviewing our paper as well as their valuable comments. We have revised the paper taking into account the comments provided. We are now submitting the revised manuscript for review and potential publication.

Response to reviewer #3

1. A thorough revision in English is necessary.

RE: Done. We have revised and corrected grammatical and scientific mistakes.

2. Most of the figures should be redrawn by enlarging labels.RE: Done. Per the reviewer suggestion, we have added more literature in Introduction and Results section about growth mechanism by CBD and chalcogenide semiconductors.

RE: Done. Per the reviewer suggestion, we have improved the SEM images and the text in the graphics.

Additionally all sections were restructured according reviewers comments and we actualized the references that give support our research.

Note: Changes to the manuscript have been marked in yellow.

Round 2

Reviewer 1 Report

I accept the manuscript.

Author Response

Response to referees for Ms. Ref. No.:  1936969

Dear Reviewers: We sincerely thank the reviewers for the time invested in reviewing our paper as well as their valuable comments. We have revised the paper taking into account the comments provided. We are now submitting the revised manuscript for review and potential publication.

Response to reviewer #1

  1. I accept the manuscript.RE: Done. We have revised and corrected grammatical and scientific mistakes.

RE: Thank you for your time in revised the manuscript.

Additionally all sections were restructured according reviewers comments and was revised in language.

Note: Changes to the manuscript have been marked in yellow.

Reviewer 2 Report

Still the main problems of the manuscript remains.

There is no description of the  difference between the present manuscript and the plethora of previous works. Moreover there is no any description of the strategy of the preparation method and what are the expected benefits. The manuscript is just a description of a classical preparation with good characterization. For these reasons I consider that the novelty of the manuscript is limited and I propose the rejection of the manuscript.

Author Response

Response to referees for Ms. Ref. No.:  1936969

Dear Reviewers: We sincerely thank the reviewers for the time invested in reviewing our paper as well as their valuable comments. We have revised the paper taking into account the comments provided. We are now submitting the revised manuscript for review and potential publication.

Response to reviewer #2

  1. Still the main problems of the manuscript remains.

There is no description of the  difference between the present manuscript and the plethora of previous works. Moreover there is no any description of the strategy of the preparation method and what are the expected benefits. The manuscript is just a description of a classical preparation with good characterization. For these reasons I consider that the novelty of the manuscript is limited and I propose the rejection of the manuscript.

RE:  In last revision, we added information about new chemical mechanism proposed to grow Bi2S3 by chemical bath deposition, we have given details about experimental part as reviewer suggested, also we added information about the benefits to obtain Bi2S3 and as this material can be applied in optoelectronic devices according with the properties obtained in the present research. We have revised and corrected grammatical and scientific mistakes.

Additionally all sections were restructured according reviewers comments and was revised in language.

Note: Changes to the manuscript have been marked in yellow.

Reviewer 3 Report

(a) The manuscript should be thoroughly revised in language.

(b) Equations/formulas should be numbered.

(c) Figures with too small labels should be redrawn by enlarging them.

(d) Labels of SEM images should be put inside the panels.

Author Response

Response to referees for Ms. Ref. No.:  1936969

Dear Reviewers: We sincerely thank the reviewers for the time invested in reviewing our paper as well as their valuable comments. We have revised the paper taking into account the comments provided. We are now submitting the revised manuscript for review and potential publication.

Response to reviewer #3

  1. The manuscript should be thoroughly revised in language.

RE: Done.

  1. Equations/formulas should be numbered.

RE: Done.

  1. Figures with too small labels should be redrawn by enlarging them.

RE: Done.

  1. Labels of SEM images should be put inside the panels.

RE: Done.

Additionally all sections were restructured according reviewers comments and was revised in language.

Note: Changes to the manuscript have been marked in yellow.

Round 3

Reviewer 3 Report

The manuscript is recommended for publication, while the quality of some figures can be further improved.

Author Response

Dear Reviewers: We sincerely thank the reviewers for the time invested in reviewing our paper as well as their valuable comments. We have revised the paper taking into account the comments provided. We are now submitting the revised manuscript for review and potential publication.
Response to reviewer #3 Comments and Suggestions for Authors 1.

The manuscript is recommended for publication, while the quality of some figures can be further improved

RE: Done.
Note: Changes to the manuscript have been marked in yellow.
